# Spontaneous Mutation Rates and Spectra of Respiratory-Deficient Yeast

**DOI:** 10.3390/biom13030501

**Published:** 2023-03-09

**Authors:** Xinyu Tu, Fan Wang, Gianni Liti, Michael Breitenbach, Jia-Xing Yue, Jing Li

**Affiliations:** 1State Key Laboratory of Oncology in South China, Collaborative Innovation Center for Cancer Medicine, Guangdong Key Laboratory of Nasopharyngeal Carcinoma Diagnosis and Therapy, Sun Yat-sen University Cancer Center, Guangzhou 510060, China; 2IRCAN, INSERM, Université Côte d’Azur, 06107 Nice, France; 3Department of Biosciences, University of Salzburg, 5020 Salzburg, Austria

**Keywords:** respiratory deficiency, mtDNA loss, mutation rate, genome instability, *MRPL25*, *ATP3*

## Abstract

The yeast petite mutant was first discovered in the yeast *Saccharomyces cerevisiae*, which shows growth stress due to defects in genes encoding the respiratory chain. In a previous study, we described that deletion of the nuclear-encoded gene *MRPL25* leads to mitochondrial genome (mtDNA) loss and the petite phenotype, which can be rescued by acquiring *ATP3* mutations. The *mrpl25Δ* strain showed an elevated SNV (single nucleotide variant) rate, suggesting genome instability occurred during the crisis of mtDNA loss. However, the genome-wide mutation landscape and mutational signatures of mitochondrial dysfunction are unknown. In this study we profiled the mutation spectra in yeast strains with the genotype combination of *MRPL25* and *ATP3* in their wildtype and mutated status, along with the wildtype and cytoplasmic petite rho0 strains as controls. In addition to the previously described elevated SNV rate, we found the INDEL (insertion/deletion) rate also increased in the *mrpl25Δ* strain, reinforcing the occurrence of genome instability. Notably, although both are petites, the *mrpl25Δ* and rho0 strains exhibited different INDEL rates and transition/transversion ratios, suggesting differences in the mutational signatures underlying these two types of petites. Interestingly, the petite-related mutagenesis effect disappeared when *ATP3* suppressor mutations were acquired, suggesting a cost-effective mechanism for restoring both fitness and genome stability. Taken together, we present an unbiased genome-wide characterization of the mutation rates and spectra of yeast strains with respiratory deficiency, which provides valuable insights into the impact of respiratory deficiency on genome instability.

## 1. Introduction

Mitochondria are the energy house of the eukaryotic cell, and generate adenosine triphosphate (ATP) to support cellular processes and functions. Mitochondrial dysfunction is associated with many disease-related defects, especially in brain and muscle tissue, where energy metabolism is highly active [1]. Most genes encoding for mitochondrial proteins reside in the nuclear genome, however, a core set of genes involved in respiration and oxidative phosphorylation are located in the mitochondrial genome (mtDNA) [2]. The mtDNA is conserved between humans and the unicellular model organism, the budding yeast *Saccharomyces cerevisiae*. Six out of eight protein-coding genes in the yeast mtDNA have homologs in the human mtDNA, including genes encoding for subunits I, II, and III of cytochrome c oxidase (*COX1*, *COX2*, *COX3*), apocytochrome b (*COB*) and subunits 6 and 8 of ATP synthase (*ATP6*, *ATP8*) [3]. Aside from protein coding genes, the numbers of tRNAs and rRNAs in the mtDNA are also similar. Moreover, the functions of mitochondrial genes involved in many cellular processes are also shared so that much of our current knowledge on mitochondrial dysfunction and associated human disorders comes from studies using *S. cerevisiae* [3,4,5]. The well-characterized genome and the rich cellular and molecular manipulation toolkits of the budding yeast makes it a key model to dissect and understand the functions of the mtDNA [3].

The mtDNA in *S. cerevisiae* lab strains (e.g., S288C) is not stable [6,7,8,9]. The absence (rho0) of mtDNA or the accumulation of mutations (rho^−^) in mtDNA can lead to a special phenotype, known as “petite”. Such strains are phenotypically characterized by defects in the respiratory chain, slow growth in fermentable carbon sources (e.g., glucose), and the inability to grow in non-fermentable carbon sources (e.g., glycerol). In addition to the loss of or mutations in mtDNA (i.e., cytoplasmic petite), mutations occurring in the nuclear genome can also generate the petite phenotype (i.e., nuclear petite) if the nuclear genes required for respiratory growth or encoding mitochondrial proteins are disrupted. Moreover, mtDNA loss is lethal in the wildtype yeast species *Kluyveromyces lactis* (petite-negative) [10]. Thus, the petite phenotype reflects stresses that cells have to deal with when respiratory dysfunction occurs. The petite cells are able to evolve by acquiring suppressor mutations such as *ATP1, ATP2*, *ATP3* (respectively encoding α, β, and γ subunit of the F1 sector in F1F0 ATP synthase) and *SIT4* (encoding a phosphatase that regulates the subunit Atp2p of F1F0 ATP synthase) that alter the function of ATP synthase and restore growth [11,12,13,14,15,16,17].

In our previous paper, we described that deletion of the *MRPL25* gene (also known as *AFO1*; located on chromosome VII), which encodes the large subunit of mitochondrial ribosomes, can result in mtDNA loss and reveal the petite phenotype. The growth defects of the *mrpl25Δ* mutant can be rapidly rescued by acquiring the *atp3^G348T^* suppressor mutation [18]. The energy charge and ATPase activity were almost the same between the *mrpl25Δ* mutant and the *mrpl25Δ atp3^G348T^* double mutant, indicating that these factors are not causal for growth restoration [18]. Instead, using mutation accumulation experiments, we observed an elevated SNV rate in the *mrpl25Δ* strain, suggesting the potential occurrence of genome instability [18]. However, the genome-wide mutation landscape and mutational signatures of the *mrpl25Δ* strain remain to be characterized for a better understanding of its genome instability. In this study, we analyzed the whole genome sequencing data from our previous mutation accumulation lines (MALs) [18], including the MALs of yeast strains with different presence–absence combinations of the *mrpl25Δ* and *atp3^G348T^* mutations as well as the wildtype and rho0 (cytoplasmic petite) control strains, to systematically characterize their respective mutation landscapes. We compared their mutation rates and spectra in terms of SNVs, INDELs, segmental and chromosomal copy number variants (CNVs and aneuploidies, respectively) and found both shared and background-specific mutational patterns for strains with respiratory dysfunction. Our findings shed light on the mutagenic effects brought about by respiratory dysfunction and provide insights into better understanding the cross-talks between nuclear and mitochondrial genome stability.

## 2. Materials and Methods

### 2.1. Strains

The *S. cerevisiae* strain C+ (*MAT alpha*) was treated with ethidium bromide to generate C+ rho0, in which the absence of mtDNA was validated by both DAPI staining and sequencing. The diploid strain JS760 was derived from a cross between C+ *mrpl25Δ* (this strain rapidly acquired *atp3^G348T^*) and C+ *MAT a*. The four monosporic haploids JS760-6A, JS760-6B, JS760-6C and JS760-6D were isolated by micromanipulation of an ascus from JS760, which have the genotype combination of *MRPL25* and *ATP3* in their wildtype and mutated status. The genotype information of these strains was compiled in Table 1. The technical details of strain construction have been described in our previous paper [18].

### 2.2. Mutation Accumulation Lines (MALs)

The six strains described above (JS760-6A, JS760-6B, JS760-6C, JS760-6D, C+, and C+ rho0) were used for our mutation accumulation experiments [18]. Each strain was propagated on YPD plates (MP, 114001222, 2% peptone, 1% yeast extract, 2% glucose, 2% agar) with four replicates. To keep the number of cell divisions between bottlenecks consistent across different strains, the normal-growing strains (JS760-6C, JS760-6D, and C+) were plated for transfer every two days and the slow-growing strains (JS760-6A, JS760-6B, and C+ rho0) were transferred every four days. In total, we accomplished 120 bottlenecks for the normal growers and 60 bottlenecks for the slow growers. The total number of cell divisions throughout the entire mutation accumulation process was approximately 2520 in the normal-growing strains and 1260 in the slow-growing strains [18].

### 2.3. Whole-Genome Sequencing and Data Analysis

Genomic DNA was extracted from the six ancestors and 24 MAL end-point clones by the Yeast Master Pure kit (Epicenter, Madison, WI, USA). All samples were sequenced using the Illumina HiSeq 4000 PE150 platform by BGI Europe A/S (Copenhagen, Denmark) [18]. The average sequencing depth across all samples was 88×. The *S. cerevisiae* S288C genome sequence and annotation (Release R64-1-1) was used as reference genome for our downstream analysis. We applied the modules 01.Short_Read_Mapping, 02.Short_Read_SNP_INDEL_Calling, and 04.Short_Read_CNV_Calling of our in-house genome analysis pipeline (https://github.com/yjx1217/Varathon, accessed on 10 August 2022) to investigate the genome-wide mutational landscape. For SNV and INDEL calling, both GATK4 [19] and freebayes [20] were used, with their outputs further intersected to reach a consensus. A series of filters were applied to remove SNV and INDEL calls that existed in ancestor strains, were located in complex regions (e.g., subtelomeres), and had poor mapping depths (≤10). All SNVs and INDELs that passed these filters were visually checked in the Integrative Genomics Viewer (IGV) [21] and ambiguous ones were further discarded. The SNV mutational signatures were analyzed and fit to the COSMIC signatures using the R/Bioconductor MutationalPatterns package [22], in which a strict refitting function “fit_to_signatures_strict” and a cutoff of 0.004 were applied. Of note, we identified a potential cross-contamination between two replicates (replicate 1 and replicate 3) of the C+ strain MALs based on finding a high proportion of shared mutations. Therefore, we excluded the C+ replicate 3 MAL in our analysis. The copy number of the mtDNA was estimated by nuclear-normalized mapping depth. Statistical analysis in this work was carried out in R-4.2.2 (https://www.R-project.org/, accessed on 31 October 2022).

### 2.4. Variant Effect Prediction and GO Term Analysis

All IGV-verified de novo variants were fed into the Ensembl Variant Effect Predictor (VEP) [23] for variant effect prediction, during which functional impacts of all variants were scored (as “low”, “moderate”, or “high”) and genic variants were assigned to their respective genes. All genic variants were used to perform Gene Ontology (GO) term enrichment analysis via the GO Term Finder module of Saccharomyces Genome Database (SGD; https://www.yeastgenome.org, accessed on 19 September 2022). A *p* value cutoff of 0.01 was used for GO term enrichment analysis.

## 3. Results

### 3.1. Mutation Accumulation Experiment Setup

In order to systematically investigate the mutation rates and spectra of yeast strains with respiratory deficiency and the petite phenotype, we applied long-term random single-cell bottlenecks, also known as mutation accumulation lines (MALs), to a series of strains: the four monosporic strains (A, B, C, D) derived from the diploid strain JS760-6 with the genotype combination of *MRPL25* and *ATP3* in their wildtype and mutated status, as well as the positive control C+ and negative control C+ rho0 (cytoplasmic petite) (Table 1). The *MRPL25* gene encodes a protein of the large subunit of the mitochondrial ribosome, the deletion of which results in mtDNA loss, respiratory deficiency, and growth defects, such as the JS760-6B strain (*mrpl25Δ ATP3*^+^) [24]. However, when accompanied with the suppressor mutation G348T in *ATP3*, the growth of the *mrpl25Δ* mutant can be restored, such as the JS760-6D strain (*mrpl25Δ atp3*^G348T^) [17,18]. Strains with the suppressor mutation alone, such as the JS760-6A strain (*MRPL25^+^ atp3*^G348T^), rapidly revealed an enhanced number of petite colonies on the YPD plate, indicating a strong tendency towards losing mtDNA (which was confirmed by sequencing, as will be discussed further on). We performed mutation accumulation experiments with single-cell bottlenecks every two or four days, depending on the growth rate of the specific strain (See Section 2 for details) during 240 days continuous culturing, which accounted for approximately 2520 or 1260 generations of random mutation accumulation, respectively (Table 1).

### 3.2. Elevated Spontaneous SNV and INDEL Rates in the mrpl25Δ Mutant

The ancestors and end-time-point MALs of all six strains described in Table 1 were sent for whole genome sequencing to characterize their respective mutational profiles. Among these strains, the two petite strains, JS760-6B (*mrpl25Δ ATP3*^+^) and C+ rho0, revealed the highest SNV rates (2.70 × 10^−9^ and 2.48 × 10^−9^ per base per generation, respectively), which are significantly higher when compared with the wildtype C+ control (1.15 × 10^−9^ per base per generation, *t*-test, *p* < 0.05) (Figure 1a, Appendix A). The more than 2-fold increases in genome-wide SNV rate revealed elevated genome instability associated with mtDNA loss. Such genome instability of petite strains was also reflected by the high INDEL rate of JS760-6B (4.27 × 10^−10^ per base per generation), although the C+ rho0 strain appeared to have a normal INDEL rate (Figure 1b, Appendix A). Interestingly, the double mutant JS760-6D (*mrpl25Δ atp3*^G348T^) showed a comparable SNV rate (*t*-test, *p* > 0.27) but significantly higher INDEL rate (*t*-test, *p* = 0.04) relative to the C+ control, indicating that the genome instability triggered by *mrpl25Δ* can only be partially compensated by the *atp3*^G348T^ suppressor mutation. The INDEL rate of the double mutant JS760-6D (*mrpl25Δ atp3*^G348T^) was on a par with that of JS760-6B (*mrpl25Δ ATP3^+^*) (*t*-test, *p* = 0.30; Figure 1b and Appendix A). Both of them showed higher INDEL rates than the other four strains without *mrpl25Δ*, suggesting that *mrpl25Δ* plays a role in the increase in INDEL mutation rate, although the underlying mechanism is unclear. This also explains the higher INDEL rates that we observed in JS760-6B (with *mrpl25Δ*) but not its C+ rho0 control (without *mrpl25Δ*).

### 3.3. Different Mutation Spectra Underlying the mrpl25Δ and rho0 Petite Strains

Next, we investigated the base substitution spectrum of SNVs (Figure 2a,b). Although the proportion of each individual substitution type does not vary significantly among the six strains (Figure 2a, chi-square test, *p* = 0.90), we observed notable differences when combining different substitution types for calculating the transition/transversion (Ts/Tv) ratio. Ts/Tv is expected to be 0.5 under the neutral model of evolution, but in reality Ts/Tv is often greater than 0.5 [6,25,26], which was also the case in this study. Notably, we observed a strong Ts/Tv bias in the C+ rho0 petite strain (mean Ts/Tv = 1.25; *t*-test, *p* = 0.002 compared with the C+ control; Appendix A), which is largely driven by the high proportion of C > T transition (41%) (Figure 2a, b). Interestingly, such a strong Ts/Tv bias was not observed in the *mrpl25Δ* petite strain (Ts/Tv = 0.89, *t*-test, *p* = 0.33 compared with the C+ control; Appendix A), reflecting different mutational signatures of these two types of petite strains. Yeast is known to have a guanine/cytosine (G/C) to adenine/thymine (A/T) mutational bias [6,27]. This was also observed in our experiments, with the number of G/C to A/T mutations relative to the number of A/T to G/C mutations ranging from 2.14 to 3.12 (Figure 2b and Appendix A). The extent of such bias was similar among the six strains (*t*-test, *p* > 0.05). We further analyzed the SNVs based on the relative incidences of base substitution within a trinucleotide context and further fit the profile to the COSMIC (Catalogue of Somatic Mutations in Cancer) signatures that characterize mutational patterns derived from cancer (Figure 2c,d). The mutational signature of C+ recaptured what has been reported in the wildtype diploid yeast strain [28], suggesting the effectiveness of our signature identification pipeline. Using this approach, we found shared mutational signatures of polymerase eta (Pol η) activity in *mrpl25Δ* and rho0 strains (SBS9, single base substitutions signature, Figure 2d). Although the cancer signatures identified in humans are not completely applicable in yeast and Pol η is not the only error-prone polymerase in humans, our results provide empirical clues for future functional studies on the role of error-prone polymerases in driving the genome instability of strains with mitochondrial dysfunction. We also identified other signatures that were not completely shared between *mrpl25Δ* and rho0 strains. The etiology of these signatures was not clearly defined by COSMIC and remains to be further investigated.

### 3.4. No Sign of Selection during Mutation Accumulation

The biggest advantage of using a mutation accumulation experiment to study the mutational process lies in its non-selective nature, which allows for a random and unbiased accumulation of mutations. In this study, we analyzed the fractions of non-synonymous SNVs and genic SNVs relative to total SNVs, which can be used as proxies to check if any sign of unintended selection was introduced during our experiment. In the model of neutral evolution, the fractions of non-synonymous SNVs and genic SNVs relative to total SNVs are expected to be 0.76 and 0.74, respectively [26]. By comparing to this neutral expectation, we found no sign of selection in any of the six strains, verifying the nature of random mutation accumulation in our experiments (Figure 3a,b, Fisher’s exact test, *p* > 0.37). We also checked if there was any functional enrichment of genic SNVs found in our MALs, e.g., beneficial mutations in F1-ATPase would largely interrupt the neutrality of the mutation accumulation system. No such enrichment was identified except that the JS760-6A strain (*MRPL25^+^ atp3^G348T^*) showed an enrichment of mutated genes responsible for ATP-dependent activity and ATP-binding cassette transporter activity. In addition, we also examined the number of SNVs distributed among different chromosomes and found a strong linear correlation regarding chromosome lengths for all six strains (Figure 4, *R*^2^ > 0.61, *p* < 3.85 × 10^−4^). This again reflects the randomness of mutations accumulated in our experiment.

### 3.5. The mtDNA Copy Number Variation and mtDNA Mutations

For the strains that initially had mtDNA (C+, JS760-6A and JS760-6C), we first analyzed the copy numbers of mtDNA across their MALs (Figure 5a). All MAL replicates of JS760-6A (*MRPL25^+^ atp3*^G348T^) lost mtDNA; this agrees with our observation that the petite phenotype appeared quickly in these lines soon after the first few transfers. Such rapid loss of mtDNA suggests a substantial mitochondrial genome instability in strains with the *atp3*^G348T^ mutation. For wild type strains, the C+ control maintained mtDNA in all MAL replicates, while the JS760-6C strain (*MRPL25^+^ ATP3^+^*) lost its mtDNA in two MAL replicates and showed the petite phenotype accordingly. On average, the MALs that maintained the mtDNA had 14 mitochondrial genomes per haploid nuclear genome.

Next, we examined the mitochondrial SNV and INDEL rates of these strains. We found the C+ and JS760-6C (*MRPL25^+^ ATP3^+^*) strains showed similar mutation rates in terms of both SNV and INDEL (*t*-test, *p* > 0.26), with an average mitochondrial SNV rate of 8.33 × 10^−9^ per base per generation (Figure 5b, Appendix A) and mitochondrial INDEL rate of 1.67 × 10^−8^ per base per generation (Figure 5c, Appendix A). Our estimates of the mitochondrial SNV rate were close to previous estimates using MALs of haploid yeast strains (12.2 × 10^−9^ by Lynch et al. [6]; 4.82 × 10^−9^ by Sharp et al. [27]), whereas our estimates of the mitochondrial INDEL rate were notably higher in comparison (7.48 × 10^−9^ by Lynch et al. [6]; 5.71 × 10^−9^ by Sharp et al. [27]). The background genetic differences of strains used across these three studies may have played a role in explaining this disparity as previously acknowledged [29].

### 3.6. Aneuploidies and Segmental Copy Number Variation

In addition to SNVs and INDELs, we also analyzed whole-chromosome (i.e., aneuploidy) and segmental copy number variants (CNVs) across all the MALs. We found the chromosome I (chrI) and XVI (chrXVI) aneuploidies existed in the JS760-6A (*MRPL25^+^ atp3*^G348T^) and JS760-6B (*mrpl25Δ ATP3*^+^) ancestor strains (Figure 6a). This was unknown when we started the mutation accumulation experiment and was likely introduced during strain construction but before spore dissection given the 2:2 segregation ratios among JS760-6A, JS760-6B, JS760-6C, and JS760-6D. These aneuploidies may respond to *MRPL25* deletion and the subsequent mitochondrial dysfunction, given it is known that aneuploidy can act as a transient or stable stage for survival or adaptation when cells are under stress [30,31,32,33,34,35]. During mutation accumulation, the chrI aneuploidy was kept to the end time point in all eight MALs, while the chrXVI aneuploidy was lost in three out of eight MALs. We also noticed that one JS760-6B MAL acquired two additional copies of chrI (a total of three copies of chrI) and one C+ MAL gained an extra copy of chrI. Aneuploidy has been reported to influence mutation rates [36]. In our system, we found that chrI and chrXVI aneuploidies do not affect our conclusion of the impact of *mrpl25Δ* on SNV and INDEL rates because both JS760-6A (*MRPL25^+^ atp3^G348T^*) and JS760-6B (*mrpl25Δ ATP3^+^*) were chrI and chrXVI aneuploidies, but they showed distinct SNV and INDEL rates, indicating that aneuploidy was not the primary driver for genome instability. Moreover, the rho0 strain was euploid and showed a SNV rate as high as the JS760-6B strain (*mrpl25Δ ATP3^+^*), suggesting mitochondrial dysfunction should have much larger effects on genome instability than aneuploidy. In addition to aneuploidy, we also analyzed segmental CNVs. Two such cases with convincing signals were presented (Figure 6b). The break points of both cases occurred in repeat regions such as long-terminal repeats (LTRs) and full-length Ty1 retrotransposable elements, which is consistent with a previous report on the strong association of genome rearrangement breakpoints and repetitive sequences in yeast genomes [37].

## 4. Discussion

The loss of mtDNA leads to a cellular crisis in yeast cells [38]. The most obvious phenotype reflecting this crisis is slow growth (petite) in YPD. On the quest for strategies to overcome such a growth defect, evolution commences and suppressor mutations such as those in *ATP3* can be acquired [17,18]. We sought to understand the mechanism of *ATP3* mutation suppression in terms of increased mitochondrial membrane potential caused by enhanced ATPase activity [38,39]. However, careful measurement did not reveal an increase in the JS760-6D strain (*mrpl25Δ atp3^G348T^*) and thus we excluded this possibility [18]. Alternatively, one evolutionary strategy of quick adaptation is to induce genome instability and thereby increase the chance of gaining beneficial mutations. After acquiring such suppressor mutations, the growth advantage of the corresponding clone can outcompete its peers and rapidly take over the population. To test the mutagenetic effect of mtDNA loss, we applied the mutation accumulation approach that minimizes the effect of selection to allow almost all types of mutations to accumulate in an unbiased way. With this approach, we proved that the mtDNA loss itself can indeed promote genome instability.

Mitochondrial-dysfunction-induced genome instability has been previously reported [38,40,41,42]. For example, the mutation rate of petite cells was found to be elevated by estimating the LOH (loss of heterozygosity) rate at two loci [38]. Our work reinforced this observation by providing direct evidence for a genome-wide mutation rate increase in a set of yeast strains with mitochondrial dysfunction. Such stress-induced genome instability increases the chance of acquiring beneficial mutations in contingency scenarios [17,18,43]. Both *mrpl25Δ* and rho0 petite strains showed elevated SNV mutation rates. Moreover, their mutational signatures revealed the activity of error-prone DNA polymerase during replication. This resembles the stress-induced mutagenesis (SIM) observed in bacteria (e.g., via SOS pathway) [44], suggesting a potentially similar mechanism. Furthermore, we found once the petite cells acquired the *ATP3* suppressor mutation that conferred a growth advantage (e.g., JS760-6D: *mrpl25Δ atp3*^G348T^), their mutation rates reversed to normal. Although the mechanism remains unclear, this observation hints that the mtDNA-loss-induced mutagenesis can be sophisticatedly regulated, which effectively reduces the evolutionary cost when growth disadvantage has been rescued and a high mutation rate is no longer needed. The mtDNA loss can lead to an insufficient supply of iron-sulfur cluster (ISC) and certain types of amino acids that are required for genome integrity, while the *ATP3* suppressor mutation can alleviate the perturbation in iron metabolism and restore the function of mitochondrial tricarboxylic acid cycle [17,38]. This could release signals to downregulate the mutagenesis activity. Although the mechanism of such mutation rate regulation needs further investigation, it is an effective strategy to deal with an acute cellular crisis. The *ATP3* mutant without *MRPL25* loss (e.g., JS760-6A: *MRPL25^+^ atp3*^G348T^) revealed a high tendency of losing mtDNA (4/4 MALs became petite) and moderately increased the SNV rate, indicating there are both genotypic and phenotypic costs of *ATP3* mutations. Taken together, the mechanisms underlying the mutagenesis regulation in petites could involve multiple molecular and cellular processes, which need further investigation.

## 5. Conclusions

We systematically examined the mutation rates and spectra of petite strains in the context of *MRPL25* and *ATP3* mutations by long-term mutation accumulation experiments. We found elevated mutation rates in strains with respiratory deficiency, which further revealed the mutational signatures of error-prone DNA polymerase activity probably contributing to the genome instability. We also found once the respiratory-deficient strain acquired a suppressor mutation to recover its fitness, its mutation rate returned to normal, hinting at delicate and effective molecular machinery regulating the mutation rate. Overall, our study provides a genome-wide mutational landscape for yeast cells with mitochondrial dysfunction, and deepens the understanding of respiratory deficiency and mtDNA loss on genome instability.

## Figures and Tables

**Figure 1 biomolecules-13-00501-f001:**
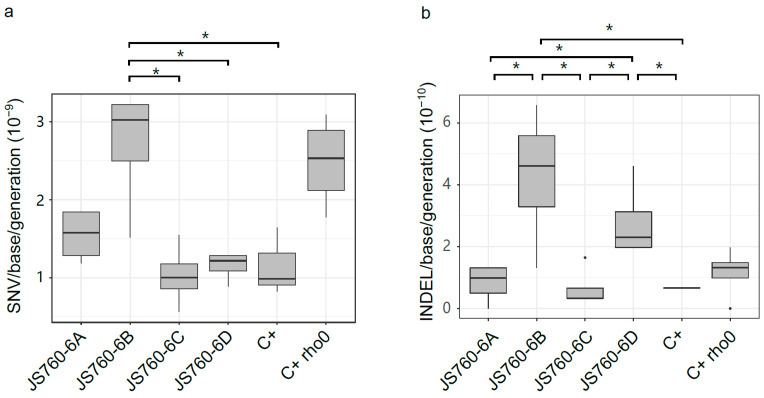
Mutation rates among the six strains. (**a**) SNV rate. (**b**) INDEL rate. The “*” symbols in the panels a and b indicate statistical significance with *t* test *p* < 0.05. The genotypes of the six strains are as follows: JS760-6A (*MRPL25^+^ atp3*^G348T^), JS760-6B (*mrpl25Δ ATP3*^+^), JS760-6C (*MRPL25^+^ ATP3^+^*), JS760-6D (*mrpl25Δ atp3*^G348T^), C+ (positive control), C+ rho0 (negative control).

**Figure 2 biomolecules-13-00501-f002:**
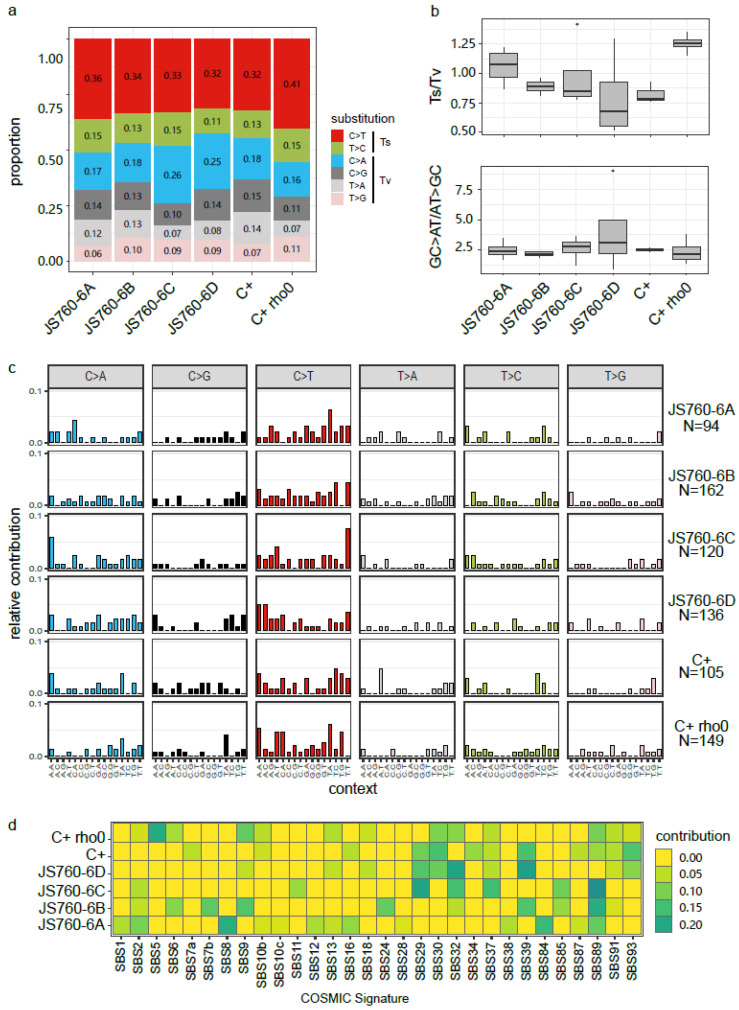
Mutation spectra comparison among the six strains. (**a**) Proportion of six types of substitutions. (**b**) Transition/Transversion ratio and G/C to A/T substitution bias. (**c**) SNV profiles within a trinucleotide context. The number of SNVs used for the analysis are 94, 162, 120, 136, 105, and 149 for JS760-6A, 6B, 6C, 6D, C+, and C+ rho, respectively. (**d**) The SNV profiles were fit to the COSMIC signature (https://cancer.sanger.ac.uk/cosmic/signatures, accessed on 13 December 2022) and the relative contributions of SBS (single base substitution) signatures are shown. The genotypes of the six strains are as follows: JS760-6A (*MRPL25^+^ atp3*^G348T^), JS760-6B (*mrpl25Δ ATP3*^+^), JS760-6C (*MRPL25^+^ ATP3^+^*), JS760-6D (*mrpl25Δ atp3*^G348T^), C+ (positive control), C+ rho0 (negative control).

**Figure 3 biomolecules-13-00501-f003:**
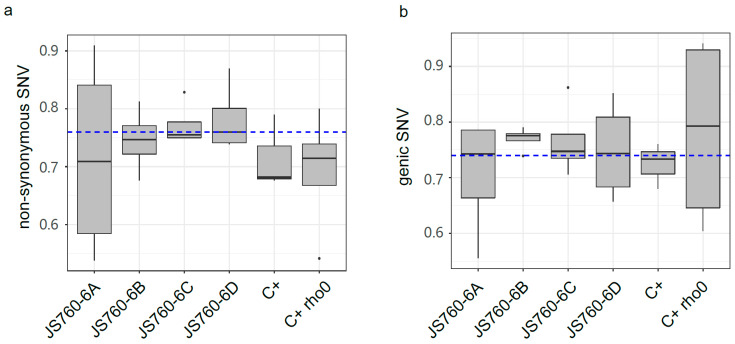
Non-synonymous and genic SNVs of the six strains. (**a**) Proportion of non-synonymous SNVs. (**b**) Proportion of genic SNVs. The blue lines in panels a and b represent the expected values for the proportion of non-synonymous SNVs and genic SNVs, respectively, under the neutral evolution model. The genotypes of the six strains are as follows: JS760-6A (*MRPL25^+^ atp3*^G348T^), JS760-6B (*mrpl25Δ ATP3*^+^), JS760-6C (*MRPL25^+^ ATP3^+^*), JS760-6D (*mrpl25Δ atp3*^G348T^), C+ (positive control), C+ rho0 (negative control).

**Figure 4 biomolecules-13-00501-f004:**
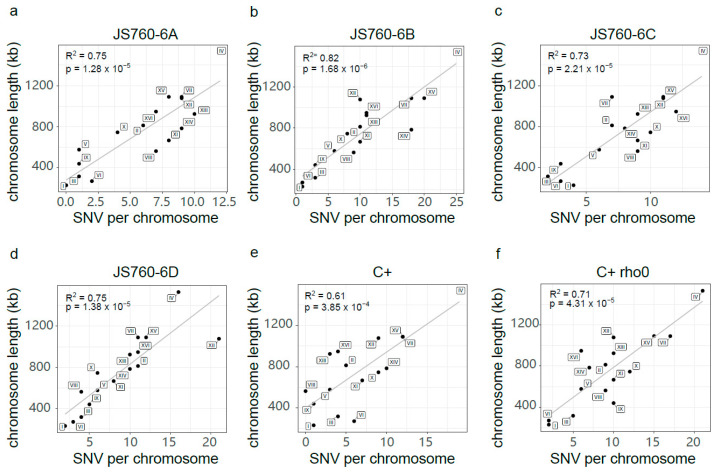
The number of SNVs positively correlates with chromosome lengths. The panels (**a**–**f**) show strong linear correlation between the number of SNVs identified on each chromosome for each strain (*x*-axis) and chromosome length (*y*-axis). The genotypes of the six strains are as below: JS760-6A (*MRPL25^+^ atp3*^G348T^), JS760-6B (*mrpl25Δ ATP3*^+^), JS760-6C (*MRPL25^+^ ATP3^+^*), JS760-6D (*mrpl25Δ atp3*^G348T^), C+ (positive control), C+ rho0 (negative control).

**Figure 5 biomolecules-13-00501-f005:**
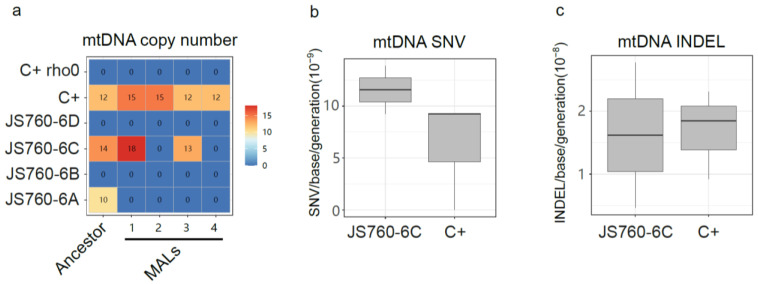
The copy number change and mutations of the mitochondrial genome. (**a**) The mitochondrial copy numbers of the six ancestor strains and their four MAL replicates. (**b**) The mitochondrial SNV rate. (**c**) The mitochondrial INDEL rate. The genotypes of the six strains are as follows: JS760-6A (*MRPL25^+^ atp3*^G348T^), JS760-6B (*mrpl25Δ ATP3*^+^), JS760-6C (*MRPL25^+^ ATP3^+^*), JS760-6D (*mrpl25Δ atp3*^G348T^), C+ (positive control), C+ rho0 (negative control).

**Figure 6 biomolecules-13-00501-f006:**
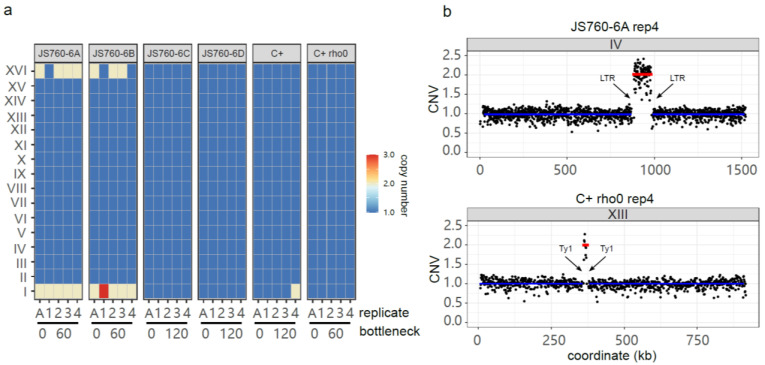
Aneuploidies and segmental copy number variants. (**a**) Chromosomal copy number changes of six ancestor stains (denoted with “A”) and their four MAL replicates. (**b**) Two cases of segmental duplication with repeat-associated breakpoints (indicated with arrows). The blue lines represent normal copy number while the red lines represent segmental duplication. The genotypes of the six strains are as follows: JS760-6A (*MRPL25^+^ atp3*^G348T^), JS760-6B (*mrpl25Δ ATP3*^+^), JS760-6C (*MRPL25^+^ ATP3^+^*), JS760-6D (*mrpl25Δ atp3*^G348T^), C+ (positive control), C+ rho0 (negative control).

**Table 1 biomolecules-13-00501-t001:** Basic information of yeast strains and their mutation accumulation lines.

Strain	Genotype	mtDNA Status	Number of Replicates	Time Interval between Bottlenecks (Day)	Total Number of Single-Cell Bottlenecks
JS760-6A	*MATa, MRPL25^+^, atp3^G348T^*	present, but tend to lose	4	4	60
JS760-6B	*MATa, mrpl25::NatMX, ATP3^+^*	absent	4	4	60
JS760-6C	*MATalpha, MRPL25^+^, ATP3^+^*	present	4	2	120
JS760-6D	*MATalpha, mrpl25::NatMX, atp3^G348T^*	absent	4	2	120
C+	*MATalpha, no auxotrophic markers*	present	4	2	120
C+ rho0	*MATalpha, no auxotrophic markers*	absent	4	4	60

## Data Availability

The sequencing data of the ancestors and MALs are available in the SRA (Sequence Read Archive) database under the BioProject ID of PRJNA632985. The key raw data are uploaded to the Research Deposit public platform (www.researchdata.org.cn, accessed on 5 February 2023), with the approval RDD number of RDDB2022390844.

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
