# Peer review of "Spontaneous Mutation Rates and Spectra of Respiratory-Deficient Yeast"

_biomolecules, 2023, doi:10.3390/biom13030501_

Round 1
Reviewer 1 Report (Previous Reviewer 1)
The manuscript has been improved, and new analyses have been added. However, a few weaknesses should be corrected.
Major. The statement about the involvement of DNA polymerase eta in the mutagenic effect by detecting SBS9 on line 237 looks light-weighted. First, the examination of Fig. 2d does not support the idea. Moreover, the signature of pol eta in human cancers may not recapitulate mutation signatures in yeast. SBS9 combines consecutive action of AID deamination (non-existent in yeast) and pol eta. If the authors insist on their statement that pol eta is involved, they should do a simple experiment with RAD30 knockout strain, where the mutator effect and signature should be gone.
Minor.
Genotypes in Table 1 instead of afo1::NourseoR it should be afo1::nat
In the Results, it will be better to mention strategic mutations than the names of strains – the reader does not have to return to Table 1.
Author Response
Please see the attached file for our point-by-point response. Thank you very much.

Reviewer 2 Report (New Reviewer)
In the manuscript entitled “Spontaneous mutation rate and spectrum of respiratory deficient yeast” the Authors performed a detailed analysis of spontaneous mutations accumulating in the WT, rho0 strains, and strains carrying mutations affecting mitochondrial function and frequently causing the mtDNA loss. The phenomenon of increased genomic instability linked to rho- or rho0 strains were already described in the literature [PMID: 19563757; PMID: 24374640.; PMID: 15126304; PMID: 21511814.] However, the analysis implemented by the Authors is well-designed and much more complex and complete than those performed earlier.
The strains construction, experiment design, as well as the data concerning the mutation frequency of strains analyzed in this work was already presented by Li et al. [PMID: 33093184 (see Mat &Meth section, Figure 9, Results and Discussion)]. In the current manuscript, instead of the mutation frequency, the Authors calculated the mutation rate (which is the more proper way to show spontaneous mutagenesis results) and added a detailed description of the sequencing results, e.g., the addition of Figure 1 b showing quantification of INDEL rate (in G3 paper the Authors only mentioned the increase in observed INDELs in mrpl25Δ strains. The Authors also made better use of acquired data by looking into mutations spectra characteristic for analyzed strains.
The changes in yeast DNA that occurred during the bottle-neck evolution experiment were also compared with human DNA sequence signatures deposited in the COSMIC database, which contains the sequence changes characteristic for different types of cancers. Based on such comparison, the Authors jumped to the conclusion that the usage of TLS DNA polymerase eta during replication is responsible for “the genome instability of mitochondrial dysfunction” (whatever it means.) They did not compare their results to the mutation spectrum revealed for the rad30Δ yeast strain, nor showed the dependence of the described mutation spectra changes on the presence of Rad30 in the cell, which means that such conclusion is an overstatement. How do the Authors know the assumed involvement of TLS DNA polymerase occurs during replication, not during gap-filling in the G2 phase? They also seem not to be aware that yeast cells contain eight DNA polymerases, while human cells possess seventeen of them. Each of them is somehow involved in TLS. Ergo, not necessarily the individual mutations signatures of DNA polymerases mutants would be the same even if the orthologues gene name matches. Moreover, the mutation spectra can vary depending on the mutation type that affected gene encoding assayed DNA polymerase. The whole gene deletion and point mutations causing amino acid substitution leading to malfunction of the catalytic domain or the other functional domain of DNA polymerase may have different mutation rates and spectra. Also, the changes in the expression level of DNA polymerases may lead to changes in the mutation rate and spectra.
The Authors did not limit their analysis to the point mutations and small INDELs, but they attempted to analyze the GRCs and aneuploidy in the set of strains they worked with. They found an aneuploidy of two chromosomes, I and XVI, in the ancestor strains. They elaborate on the further changes in chromosome numbers, which leads them to conclude that chromosome I can be prone to multiplication due to its small size and relatively small cost of maintaining it. This is a weak explanation. Aneuploidy was already shown to influence the mutation rate. Moreover, the disomy of different chromosomes causes different changes in the mutation rates and their spectrum [see the papers from Angelica Amon lab, e.g., PMID: 21852501, but also other labs’ work, e.g., PMID: 32001709] Also, it has been shown that aneuploidy may be caused by stressful conditions and act as a transient or stable stage leading to adaptation or allowing survival [PMID: 25519894; PMID: 30668788; PMID: 22286062; PMID: 23197825; PMID: 30463853] Thus, it is pretty likely that the multiplication of chromosome I was the cellular adaptation to the mitochondrial problems caused by background mutations, the lack of MRPL25 or atp3-G348T. Especially that chromosome I disomy accompanying, e.g., ULP2 loss allows rapid adaptation, but long-term adaptation restores euploidy. [PMID: 30575729].
It is unclear if the aneuploidy was considered while calculating the mutation rates.
The authors use abbreviations in the abstract, the meaning of which was explained much later in the manuscript. Not all readers have to know what SNV or INDEL means. If you do not want to lose the readers’ attention, you should explain the abbreviations at the first time you use them.
For consistency and clarity, please use the gene names, not aliases, according to nomenclature established for yeast genes by the yeast researchers community and enclosed in the Saccharomyces Genome Database, i.e., MRPL25 instead of AFO1.
Please use the mutant names according to the rules for Saccharomyces cerevisiae nomenclature, i.e., small letters italic style (e.g., atp3-G438T).
Round 2
Reviewer 2 Report (New Reviewer)
Most of my comments were addressed by the Authors. Still, they do not want to use the gene names according to SGD guidelines. They should at least mention at the beginning of the manuscript that they will use AFO1 instead of MRPL25 for this manuscript's purposes. If they think the AFO1 is the most proper name for this gene, they should write to SGD teem asking for a change in this gene cart. But they need to possess hard arguments to do that.
The Authors upgraded the notation of the AFO1 gene deletion, which they were using to a proper form afo1:: NatMX, but they consequently used the improper notation for naming the point mutation in this gene. I highly recommend the Authors become familiar with yeast S. cerevisiae nomenclature, which can be found at these two addresses: [https://www.sciencedirect.com/science/article/pii/S007668790250954X] and
[http://onlinelibrary.wiley.com/doi/10.1002/9783527636778.app1/pdf]
The Authors should also add an example from the literature, which will be proper for the statement that the stable aneuploidy stage might allow the survival of yeast cells with some mutations in their genome. It has not to be the example I know; just find another one.
Author Response
Please see the point-by-point response in the attachment. Thank you very much.

This manuscript is a resubmission of an earlier submission. The following is a list of the peer review reports and author responses from that submission.
Round 1
Reviewer 1 Report
The manuscript elaborates on the previous findings published by a partially overlapping group of authors in 3G in 2020 (ref. 10 of the current manuscript) that mitochondrial dysfunction leads to a relatively small but statistically significant increase in the accumulation of mutations in yeast. It is unclear if new experiments were conducted or if the results extend the analysis of the data obtained in the previous paper. At least, the descriptions of mutation accumulation experiments are almost identical in the two works. Therefore, we see that the novelty element in the current manuscript is incremental. The authors repeat the hypothesis presented in [10], explaining the accumulation of suppressors in mutants with mitochondrial function defect by the increased mutation rate. It needs to be clarified if the slight increase supports this claim.
The oversimplified analysis of mutations accumulated in the studied group of strains shows some differences but does not give clues to finding the mechanism of elevated mutation rates. We do not see attempts to use modern Alexandrov-Stratton technology to find the DNA sequence context of mutations and, thus, genuine signatures of mutation processes operating when mitochondrial function is down.
Other comments.
Lines 13-14. Yeast is a facultative anaerobe, so the first phrase in the Abstract looks imprecise.
Line 15. ”…somehow rescued…” looks weak, especially in the Abstract.
Line 19. “…tetratypes…” in the Abstract could be misleading for many potential readers.
Lines 21-22. Can the authors be more precise about what was seen and what “hints” were obtained?
Line 38. How much is it conserved?
Lines 85-87. Something is grammatically wrong in this sentence.
Lines 80-87. The authors should refer to {10] instead of attempting to describe strains again.
Lines 94-105. The text is similar to ref [10].
Line 170. It should be “base substitution SNVs spectrum” instead of “substitutional”
Lines 179-180. Weak and diffuse statement.
Fig 1c. Are these differences significant? If yes, what does this mean?
Lines 232-233. Isn’t it widely accepted that strain background is important?
Lines 282-283. The resemblance to stress-induced mutagenesis is slim. Did the authors see any evidence of SOS-like mechanisms?
Line 285. Weak and diffuse statement.
Line 290. Weak and diffuse statement.
Line 292. Did the authors consider that depletion of iron-cluster leads to defects of many DNA replication and repair proteins, and this is a cause of genomic instability? (e.g., Nucleic Acids Res. 2021 Jun 4;49(10):5623-563)
Lines 295-296. We agree that the nature of the signals is elusive, and further investigation is required.
Reviewer 2 Report
The authors have previously shown that loss of mtDNA in the yeast afo1 mutant defective in mitochondrial protein translation leads to mitochondrial DNA (mtDNA) loss, and that the growth defect of the cells can be rescued by the G348T mutation in ATP3 gene encoding the gamma-subunit of F1-ATPase. In the current study, they showed that afo1Δ and rho0 cells have increased mutation rates in the nucleus. The petite-related increase in nuclear mutations is suppressed by the ATP3 mutation. They cultured rho0 and rho+ cell with or without atp3(G348T) and determined mutation accumulation in the nuclear genome using Next-gen sequencing technology. They found that afo1Δ and rho0 cells have increased SNVs. This phenotype is suppressed in afo1Δ cells expressing atp3(G348T). Although afo1Δ cells have increased INDELs, this is not seen in rho0 cells. Further analysis of the data suggested that nuclear mutations accumulate in afo1Δ and rho0 cells randomly and there is no selection involved. The authors concluded that increased nuclear mutation is an evolutionary strategy to gain mutations in genes such as ATP3 for adaptation to the severe growth defect in rho-zero cells. Although the sequencing data are interesting, I have the following major concerns.
MAJOR CONCERNS:
(1) If increased nuclear mutation is an adaptive strategy in rho zero cells, it would be expected that mutations in the F1-ATPase genes should be enriched. It is surprising that there is no mention about this.
(2) The authors speculated that the primary stress from rho zero cells is a deficiency in iron sulfur cluster biosynthesis. If this is the case, any explanation for the suppression by the atp3 mutation? Ample evidence has been published in the literature that support a role of mitochondrial membrane potential in sustaining the viability of rho-zero cells. This is missed in this manuscript. Increased mtDNA instability in atp3 mutant has been reported by several groups. The authors cannot ignore this.
(3) Likewise, the authors failed to cite correctly the published literatures. For instance, references 8 and 9 rediscovered the F1-ATPase mutations that were first published by Chen and Clark-Walker in 1990s. The authors should also be aware of the original discovery that mtDNA-encoded OXPHOS genes are essential for cell survival (https://pubmed.ncbi.nlm.nih.gov/8917319/). Line 51: no mention of how SIT4 is involved.
(4) The authors concluded that “…it is afo1Δ rather than mtDNA loss or being petite that causes the increase of INDEL mutation rate” - afo1Δ cells are supposed to be rho-zero. It is difficulty to reconcile the difference between these strains. Spontaneous point and INDEL mutations may just need different time scale to occur, instead of involve different stressors.
(5) Complete loss of mtDNA needs to be confirmed in afo1Δ cells.
Minor points:
· The full genotype of yeast strains should be included in the manuscript. Please follow the standard genetic nomenclature for yeast.
· The annotations in the figures are too small to read.
· It is surprising that the growth medium was not described in the manuscript in the preparation of MALs. I assume that cells were grown in YPD, which should be clearly indicated.
· Line 33: “to support” instead of “to supply”
· Line 41: “functional mechanisms”, please reword.
· Fig. 1a: Why the p-value is not annotated between JS760-6a and 6B? I understand that -6A has lost mtDNA. This should be described early in the text to avoid confusion.